# Unify in Isolation: Unified Data Synthesis for Multi-Stage Recommendation Systems

## Abstract

As modern information processing systems grow increasingly complex, multi-stage composite frameworks are receiving heightened attention, as models with divergent optimization objectives are able to extract more heterogeneous information. A prominent and widely adopted multi-stage framework in industry is the multi-stage recommendation network, comprising three sequential stages: Recall, Coarse Ranking, and Fine Ranking. Inspired by the data-centric paradigm, we seek to develop a unified data synthesis framework applicable across diverse training objectives. Specifically, we introduce the **Uni**fied **D**ata **S**ynthesis system for multi-stage frameworks, which adaptively provides unified structured data of varying quality at different stages to consistently enhance overall recommendation data quality. Initially, UniDS utilizes Real Entropy to evaluate data quality and, through Metric-Oriented Gradient Comparison Theory, demonstrates that different stage objectives exhibit distinct sensitivity for Real Entropy. Subsequently, leveraging this difference, UniDS injects Real Entropy into user sequence segmentation via the Pattern Mining via Conditional Entropy module, aiming to mine interaction patterns among stages. Finally, UniDS establishes a unified, model-agnostic data generation architecture based on the Special Pattern-Token paradigm, which utilizes patterns separated by entropy, thereby simultaneously generating new data and core task representations. This approach ultimately achieves a unified multi-stage data generation paradigm. Extensive experiments on benchmark datasets demonstrate enhanced performance on each model in a multi-stage system, improved flexibility in feature synthesis, and superior stage adaptation. Our anonymized code is available at https://anonymous.4open.science/r/UniDS-9510/.

## 1 Introduction

A unified framework paradigm is currently a key pursuit in recommender systems. Approaches such as HSTU Zhai et al. (2024) and OneRec Zhou et al. (2025) achieve success in both academic and industrial settings by unifying models across different stages like Recall (Yu et al., 2019), Coarse Ranking (Li et al., 2022), and Fine Ranking (Fan et al., 2022), thereby alleviating problems of cascading errors and inconsistent optimisation objectives. While model unification has become an established trend, numerous studies on the scaling laws of large recommendation models have demonstrated that recommendation performance L and scaling potential depend not only on the model N but also on the data D. To improve recommendation quality at the data level, data augmentation methods such as RepPad(Dang et al., 2024b) and FRec(Nian et al., 2024) focus on enhancing data quality, while data generation methods Yin et al. (2024) have attracted attention for their ability to simultaneously increase data quantity and quality, thereby expanding recommendation capabilities. However, these data augmentation methods typically address only a single stage of the recommendation pipeline, and are thus still subject to issues of cascading errors and inconsistent optimisation objectives. In this context, Theorem 4.1 demonstrates a consistent pattern of data sensitivity across objectives, thereby facilitating the feasibility of unified data synthesis.

Overall, our work seeks to develop a unified data synthesis framework applicable across the various training objects of the multi-stage recommendation system. This approach faces two fundamental challenges: 1) Data Quality Unification. As demonstrated in Section 5.2.1, different stages of the model exhibit varying sensitivities to data quality. Within a unified framework, enhancing data quality often requires pruning redundant or insignificant information (Zhao et al., 2021), which may result in a reduced data quantity. Effectively reconciling and adaptively addressing these heterogeneous quality

requirements across stages remains a critical challenge for unified multi-stage data augmentation. 2) Data Format Unification. After aligning the quality requirements, it is essential to standardize the data generation procedure across stages. Nevertheless, each recommendation stage prioritizes different feature sets—for example, coarse ranking attends to user domain information, whereas fine ranking emphasizes behavioral data. Unifying the representation and formatting of these disparate features constitutes another major challenge in building a unified multi-stage data augmentation framework.

To address these challenges, we introduce UniDS, a **Uni**fied **D**ata **S**ynthesis system for multi-stage frameworks. Towards Challenge (1), we propose the **Metric-oriented Gradient Comparison Theory**, utilizing Real Entropy as the data evaluation metric and demonstrating that different SR stages exhibit varying sensitivities to Real Entropy. Based on this theory, we develop the **Pattern Mining via Conditional Entropy Paradigm**, which segments user sequences by computing conditional entropy values and controlling entropy thresholds to mine diverse patterns, thereby enabling adaptive and unified entropy control within user sequence data. Towards Challenge (2), we present the **Uniform Dataset Generation Paradigm**, which employs unified Pattern-Token sequence representations and a standardized mechanism for sequence and representation generation. This approach facilitates consistent formatting and flexible synthesis of features required by different recommendation stages, thereby establishing a unified data generation logic for multi-stage augmentation. Through the integration of theoretical insights and practical modules, our framework enables unified data generation across multiple recommendation stages. Our contributions are summarized as follows:

- We propose UniDS, the first unified data synthesis framework for multi-stage recommendation system, bridging the gap between coarse multi-stage practices. UniDS enables the creation of high-quality, model-agnostic data for all possible multi-stage complex structures.

- We introduce Metric-oriented Gradient Comparison Theory and a conditional entropy-based pattern mining module. Real Entropy metric serves as a theoretically-grounded unified data quality metric for analyzing stage-specific sensitivities, while pattern mining adaptively extracts meaningful patterns via conditional entropy thresholds, supporting unified and adaptive entropy control.

- We present the Uniform Dataset Generation paradigm, employing standardized Pattern-Token sequence representations and a unified generation logic for flexible and consistent feature synthesis across different stages in multi-stage complex structures.

- The effectiveness and applicability of UniDS are demonstrated both theoretically and experimentally. Theoretical analysis highlights the benefits of unified entropy control and data generation, while extensive experiments on benchmark datasets confirm improved recommendation performance, feature synthesis flexibility, and stage adaptation.

## 2 RELATED WORK

### 2.1 MULTI-STAGE COMPLEX SYSTEM

Multi-stage complex systems have developed rapidly in SR, Retrieval-Augmented Generation (RAG), and Large Language Model (LLM) agents. In multi-stage sequential recommendation, recent studies explored personalized multi-stage pipeline adaptation Ding et al. (2025) and multi-stage cascading frameworks for sequential pre-ranking Wei et al. (2024), highlighting the effectiveness of modular and progressive information refinement for practical deployments. RAG methods integrate neural retrieval and generative models, enabling language systems to incorporate external knowledge into understanding and generation. Representative works introduce RAG for knowledge-intensive tasks Lewis et al. (2020), adapt it for open-domain question answering Asai et al. (2021), and enrich retrieval diversity by mixture-of-context designs Islam et al. (2024). LLM agents have become a new paradigm for multi-stage reasoning; novel approaches synergize reasoning with interactive actions Yao et al. (2022), investigate agent evaluation across diverse setups Xi et al. (2023); Han et al. (2023), and allow models to dynamically teach themselves tool usage for downstream tasks Schick et al. (2023). These studies all demonstrate the potential of multi-stage architectures, but they continue to face the aforementioned dilemma of heterogeneous information related to diverse training objectives among models, which remains an open challenge to be addressed. Among these multi-stage frameworks, the application scenarios of sequential recommendation frameworks are more mature and widespread, and the resulting economic benefits are particularly significant. This is why we select the sequential recommendation framework to validate our multi-stage synthesis.

## 2.2 SEQUENTIAL RECOMMENDATION

Sequential recommendation (SR) models user interests from historical interactions He et al. (2023); Tong et al. (2024); Wu et al. (2024). Early methods used statistical approaches like Markov chains He & McAuley (2016) and collaborative filtering Choi et al. (2012). Deep learning brought neural models such as CNNs Tang & Wang (2018b), GNNs Wu et al. (2019); Wang et al. (2020); Wu et al. (2018), and RNNs Hidasi et al. (2015), with GRU4Rec Hidasi & Karatzoglou (2018) and Caser Tang & Wang (2018a) being notable examples. Recent advances leverage attention and Transformer models such as SASRec Kang & McAuley (2018a) and BERT4Rec Sun et al. (2019a), while large-scale models and datasets (e.g., LLaMA4Rec Luo et al. (2024), HSTU Zhai et al. (2024), OneRec Zhou et al. (2025)) improve scalability. Nevertheless, data quality is also crucial for performance. Data augmentation for SR has shifted from simple random operations—which risk losing important interactions—to more guided strategies using auxiliary information, such as item importance Wang et al. (2022), temporal signals Tian et al. (2022); Dang et al. (2023b;a), periodicity Tian et al. (2023), and user behaviors Xiao et al. (2024). Scenario-specific methods target domains like music Oh et al. (2023), trajectories Zhuang et al. (2024), baskets Li et al. (2023); Nian et al. (2024), and short sequences Dang et al. (2024b). Generative approaches, such as DR4SR Yin et al. (2024), further enhance data quality. However, most augmentation methods are tailored to specific tasks or models, complicating their application in multi-stage frameworks and sometimes causing optimization inconsistencies. Our framework addresses these challenges.

## 3 PRELIMINARY AND DEFINITION

### 3.1 PROBLEM DEFINITION

**Sequential Recommendation** SR Zhang et al. (2024); Wang et al. focuses on predicting the next item a user is likely to interact with, given a sequence of previous interactions. This task is essential in personalized systems, as it considers the temporal dynamics and evolving preferences of users. Formally, let $U$ denotes the set of users and $I$ denotes the set of items. For a specific user $u \in U$, we represent their interaction history as an ordered sequence $S_u = [i_1, i_2, \ldots, i_{|S_u|}]$, where each $i_k \in I$ indicates an item interacted with at time step $k$. The objective is to model the probability distribution over $I$, given $S_u$, to predict the subsequent item $i_{n+1}$. This problem is challenging due to the complex, dynamic patterns in user behavior, which are influenced by various contextual factors.

### 3.2 PRELIMINARY

**Real Entropy** We introduce the Real Entropy Song et al. (2010) ($S^{real}$) factor to introduce data quality among models in different stages. Real entropy is a refined measure that captures user interaction distributions across patterns of varying lengths. It is computed as: $S^{\text{real}} = -\sum_{S' \subset S_u} P(S') \log_2[P(S')]$, where $S_u$ represents each user sequence. $P(S')$ represents the probability of each subset of subsequence $S'$ By calculating the entropy of distributional differences in user interactions, we make a correction for removing redundant low-entropy item sequences in SR, ultimately addressing the uniform quality measure deficiency. However, the complexity of calculating $S^{real}$ using the above definition is extremely high. To reduce complexity, we utilize Lemma 3.1:

**Lemma 3.1.** *LZ compression Ziv & Lempel (1977). For a user interaction sequence with length $|S_u|$, its Real Entropy $S^{real}$ is estimated by: $S^{real} = \left( \frac{1}{|S_u|} \sum_j \Lambda_j \right)^{-1} \ln |S_u|$, where $\Lambda_j$ denotes the minimum length $j$ such that the subsequence starting from position $i$ with length $j$ does not appear as a continuous sub-sequence of $S_u = [i_1, i_2, \ldots, i_{|S_u|}]$.*

Real Entropy alone is insufficient to establish a direct connection with the final performance of the model. To facilitate further investigation, we propose the User Predictability Rate $\Pi_{max}$, a metric closely associated with Real Entropy:

**User Predictability Rate** The user predictability rate ($\Pi_{\max}$) Song et al. (2010) quantifies the maximal achievable accuracy in predicting a user's next location, given their historical sequence. Formally, $\Pi_{\max}$ is defined as: $\Pi_{\max} = \mathbb{E}_{h_{n-1}} [\max_x P(x|h_{n-1})]$, where $P(x|h_{n-1})$ denotes the probability of visiting location $x$ given the user history $h_{n-1}$. Given the Real Entropy $S^{\text{real}}$, $\Pi_{\max}$

satisfies the following relationship:

$$S^{real} = -(\Pi_{max} \log_2 \Pi_{max} + (1 - \Pi_{max}) \log_2(1 - \Pi_{max})) + (1 - \Pi_{max}) \log_2(|I| - 1), \quad (1)$$

where $N$ denotes the number of unique locations. A higher $\Pi_{max}$ indicates greater predictability and lower entropy in user behavior.

## 4 METHODOLOGY

Overall, we identify two principal challenges in implementing a unified data augmentation framework: data quality unification and feature format unification. To clearly illustrate the overall process, we present the complete UniDS pipeline in Figure 1. Specifically, as depicted in Part A, we first employ the Real Entropy $S^{real}$ to assess data quality, as it reflects the informational density of the data. We demonstrate that the gradient sensitivity $\frac{\delta(\cdot)}{\delta S^{real}}$ to data quality varies across distinct recommendation stages. Building on this theorem, we introduce pattern mining via a conditional entropy module, which segments user sequences according to computed conditional entropy values and extracts meaningful patterns based on entropy, as shown in Part B. Finally, we utilize unified pattern-token sequence representations and a standardized mechanism for sequence and representation generation in Part C. By integrating theoretical foundations with practical modules, our framework enables unified data generation throughout multiple recommendation stages.

### 4.1 METRIC-ORIENTED GRADIENT COMPARISON THEORY

To unify the requirements for data quality across different stages, we first synthesize the relationship between evaluation metrics at various recommendation stages and data quality. We assess data quality using the Real Entropy ($S^{real}$) metric, as described in Section 3.2. This choice is motivated by our analysis, which demonstrates that Real Entropy possesses the following desirable properties:

**Theorem 4.1.** *Assuming $rank_i$ denotes the rank assigned by the model to the $i$-th target positive item, $\overline{rank_i}$ represents the model-assigned rank for the item to be predicted at the top position, and $S^{real}$ denotes the Real Entropy of the user sequence, the sensitivity of the fine-ranking metric AUC, the coarse-ranking metric NDCG, and the recall metric MRR to Real Entropy is ordered as follows:*

$$\frac{\delta AUC}{\delta S^{real}} = -o(\frac{1}{rank_i}) \geq \frac{\delta NDCG}{\delta S^{real}} = -o(\frac{1}{rank_i \log_2(rank_i)^2}) \geq \frac{\delta MRR}{\delta S^{real}} = -o(\frac{1}{rank_i^2}) \quad (2)$$

The detailed proof of Theorem 4.1 is elaborated in Appendix A.3. At a high level, we establish our results by proving the following two key equations:

$$\delta S^{real} = -(\frac{\ln(\Pi_{max}) + 1}{\ln 2} - \frac{\ln(1 - \Pi_{max}) + 1}{\ln 2})\delta \Pi_{max} - \log_2(|I| - 1)\delta \Pi_{max}, \quad (3)$$

$$\frac{\delta AUC}{\delta S^{real}} = -o(\frac{1}{rank_i}) \frac{1}{\log_2 \frac{(1 - \Pi_{max})}{\Pi_{max}(I - 1)}} \frac{\Sigma_{i \in N^+} \overline{\delta \, rank_i}}{\delta \Pi_{max}}, \frac{\delta NDCG}{\delta S^{real}} = -o(\frac{1}{rank_i \log_2(rank_i)^2})$$

$$\frac{\Sigma_{i \in N^+} \overline{\delta \, rank_i}}{(\log_2 \frac{(1 - \Pi_{max})}{\Pi_{max}(I - 1)} \delta \Pi_{max})}, \frac{\delta MRR}{\delta S^{real}} = -o(\frac{1}{rank_i^2}) \frac{1}{\log_2 \frac{(1 - \Pi_{max})}{\Pi_{max}(I - 1)}} \frac{\Sigma_{i \in N^+} \overline{\delta \, rank_i}}{\delta \Pi_{max}},$$

$$(4)$$

where $\Pi_{max}$ denotes the maximally achievable predictability rate given the user's sequence, $|I|$ is the total number of items, $N^+$ represents the set of positive items, $rank_i$ is the model-assigned rank for positive item $i$, and $\overline{rank_i}$ is the average top-predicted rank.

Eq. 3 links Real Entropy $S^{real}$ with user predictability $\Pi_{max}$, while Eq. 4 propagates changes in predictability to metric sensitivities via $\frac{\delta(\cdot)}{\delta S^{real}}$; thus, variations in data entropy affect final ranking performance via their influence on $\Pi_{max}$. After revealing the sensitivity of Real Entropy-based data quality under different stage metrics through gradient difference analysis of Real Entropy, we propose to integrate entropy measurement into the data generation framework. We design a Pattern Mining via Conditional Entropy structure that segments user sequences $S_u = [i_1, i_2, \ldots, i_{|S_u|}]$ into entropy-aware patterns $S_u = [P_1, \ldots, P_k]$, where each pattern $P_1 = \{i_1, i_2\}$, $P_2 = \{i_3\} \ldots$ represents a segmented subsequence split by entropy and k denotes the total number of Pattern P.

Figure 1: Illustration of UniDS. Parts A, B, and C in the figure correspond to Section 4.1, Section 4.2, and Section 4.3, respectively.

## 4.2 PATTERN MINING VIA CONDITIONAL ENTROPY

Given user interaction sequences $S_u = [i_1, \ldots, i_{|S_u|}]$, our method segments these sequences into robust, low-entropy patterns by leveraging both empirical and model-derived entropy. Let $f_i$ denote the frequency of item $i$, and $F$ the total number of interactions, such that $p(i_1) = \frac{f_{i_1}}{F}$. The conditional entropy for item $i_t$ with position $t$ under an external sequential probabilistic model with distribution $p_m$ over user sequence $S_u$ is defined as:

$$S^{real} = H(S_u) = -\sum_{i_t \in S_u} p_m(i_t \mid S_u[< t]) \log p_m(i_t \mid S_u[< t]), \tag{5}$$

To extract meaningful patterns, we generate initial seeds $P = [i_1]$ for each item with $H(i_1) < \theta$ and sufficient support. For iterative extension, a pattern $P = [i_1, \ldots, i_n]$ is expanded by appending observed successor items $s$, forming $P' = [i_1, \ldots, i_n, s]$ as if they satisfy the constraints:

$$\mathcal{E} = \{P' : H(P') < \theta, \, \text{Occu}(P) \geq c, \, |P'| \leq L_{\max}\}, \tag{6}$$

where the global threshold $\theta$ is determined as a scaled mean entropy across different recommendation stages and $\text{Occu}(P)$ is the pattern's occurance count . The minimum occurrence threshold, $c$, is set to 2 to preventuser the extraction of insignificant patterns, while $L_{\max}$ denotes the maximum pattern length. By exploiting the non-negativity and additive properties of entropy, specifically $H(P') = H(P) + h(s \mid P)$, we utilize dynamic programming to cache entropies for patterns of length $L_{\max}$. This enables efficient computation of entropies for patterns of length $n+1$ by retrieving the stored value of $H(P)$ and adding the incremental entropy $h(s \mid P)$. The process is iterated until no further valid extensions remain. This framework effectively segments sequences into high-support, entropy-based tokens, resulting in subsequences that are well-suited for downstream multi-stages. To further illustrate our algorithmic details, we present the pseudocode in Appendix A.5.

Following entropy-driven pattern segmentation, we then inject auxiliary user preference as special tokens $T$ into the sequence through a unified inference paradigm, ultimately leading to a unified paradigm for multi-stage data augmentation.

## 4.3 UNIFORM DATASET GENERATION

### 4.3.1 UNIFORM DATA PRETRAINING FORMALIZATION

After obtaining multi-stage unified data requirements via the entropy-driven pattern segmentation, we unify both the data format and generative logic across the entire recommendation pipeline by enhancing the original pattern-only sequence $[P_1, P_2, \ldots, P_k]$ into an interleaved pattern-token sequence $[P_1, T_1, P_2, T_2, \ldots, P_k, T_k]$. In this unified framework, each $T_i$ functions as a special token denoting user attributes, session context, or domain-specific information, while each $P_i$ denotes an entropy-based behavioral segment derived from user interaction history. This PT-sequence structure provides a principled way to incorporate explicit Feature signals alongside fine-grained behavioral patterns, enabling the model to support flexible multi-stage reasoning and dynamically adapt to user state transitions throughout different stages of the recommendation process. By representing each segment as $\phi(T_i, P_i; \Theta)$, where $\phi$ is the target sequential model via different stages with parameters $\Theta$, We can use $T_i$ to provide different key core information to the model at different

stages of recommendation, effectively bridging context-dependent logic and behavioral modeling within the same architecture. Furthermore, this design facilitates joint pretraining over both tokens and patterns, captured by the unified objective $\mathcal{L}_{\text{pretrain}} = \sum_i [\mathcal{L}_{\text{pattern}}(P_i) + \mathcal{L}_{\text{token}}(T_i)]$, which encourages the learning of both contextual understanding and behavioral prediction. The interleaved PT-sequence format thus establishes a robust and interpretable foundation for end-to-end multi-stage recommendation, ensuring seamless integration and unified reasoning across all steps in the pipeline.

### 4.3.2 Uniform Data Generation Structure

Let the input to the regenerator be an entropy-partitioned sequence $S_u^G = [P_1, T_1, P_2, T_2, \ldots, P_k, T_k]$, where $T_i$ is the token at position $i$ (e.g., an item or domain identifier), and $P_i$ is the corresponding entropy-based pattern (subsequence segment). The model aims to generate $S_u'^G = [P_1', T_1', P_2', T_2', \ldots, P_k', T_k']$, which reconstructs or generates content aligned to each entropy segment.

First, the input sequence $S_u^G$ is encoded by a Transformer encoder to produce a global representation vector $\mathbf{m} \in \mathbb{R}^D$, where $D$ is the hidden dimension. To accommodate diverse structral interpretations in decoding, we project $\mathbf{m}$ into $K$ different semantic subspaces, yielding $\{\mathbf{m}_1', \mathbf{m}_2', \ldots, \mathbf{m}_K'\}$ with each $\mathbf{m}_k' \in \mathbb{R}^D$. This mechanism enables the decoder to access a richer set of memories, facilitating pattern diversity. For each entropy pattern segment $[T_i, P_i]$, we apply a second Transformer encoder to obtain a target pattern representation $\mathbf{h}_{\text{pattern}}^{(l)} \in \mathbb{R}^D$, summarizing the structral interpretations of the specific target pattern. To avoid trivial "copy" solutions and enforce adaptive decoding, we transform $\mathbf{h}_{\text{pattern}}^{(l)}$ into a probability vector $\boldsymbol{\pi} \in \mathbb{R}^K$ using a Multilayer Perceptron followed by softmax activation:

$$\boldsymbol{\pi} = \text{Softmax}(\text{MLP}(\mathbf{h}_{\text{pattern}}^{(l)})), \tag{7}$$

where $\pi_k$ denotes the probability of selecting the $k$-th projected memory. This step ensures that each target pattern can dynamically select a combination of structural spaces, improving pattern flexibility.

Instead of hard selection (which inhibits gradient flow and under-trains branches), we use a weighted sum $\mathbf{m}_{\text{final}} = \sum_{k=1}^K \pi_k \mathbf{m}_k'$ to aggregate all projected memories according to $\boldsymbol{\pi}$. This aggregated memory $\mathbf{m}_{\text{final}}$ (with gradients) serves as context for the decoder to autoregressively generate $[T_l', P_l']$ at each position $l$. Training employs a cross-entropy loss for reconstruction. For output tokens $\hat{x}_j$ (where $j$ indexes over the sequence of pattern tokens), we maximize the log-likelihood given previous outputs and final memory.

$$\mathcal{L}_{\text{rec}} = -\sum_{b=1}^B \sum_{j=1}^{|\mathbf{X}_b|} \log p(x_{b,j} \mid \hat{\mathbf{X}}_{b,<j}, \mathbf{m}_{b,\phi(j),\text{final}}), \tag{8}$$

where B denotes the number of training batches. To further regularize the model and encourage usage of diverse structural memories, we include an entropy regularization term on $\boldsymbol{\pi}$ with $\mathcal{L}_{\text{div}} = -\sum_{k=1}^K \pi_k \log \pi_k$ which penalizes overly confident selections and promotes diversity. The total training objective is then: $\mathcal{L} = \mathcal{L}_{\text{rec}} + \mathcal{L}_{\text{div}}$. By segmenting sequences with entropy-based patching and explicitly modulating decoder memory via a learned pattern-dependent distribution, this approach adaptively reconstructs complex user histories while enhancing semantic diversity across generated segments. Each step is designed to avoid degenerate identity mapping, encourage rich memory usage, and robustly model sequence regeneration from structured, entropy-derived input.

## 5 Experimental Evaluation

**Datasets.** To demonstrate the performance of our proposed approach across various kinds of datasets, we conducted experiments on three publicly available datasets: **MovieLens-1M** Harper & Konstan (2015) (ML-1M), **Amazon Books** McAuley & Leskovec (2013) (AMZ-Books), **KuaiRand-Pure** Gao et al. (2022) (KR-Pure). The specific details of the dataset are presented in Appendix A.1.

**Baselines.** To rigorously validate that our model consistently outperforms prevailing data augmentation methods across different frameworks, we conduct extensive comparisons between HSTU and LLaMA frameworks, evaluating both their native implementations and their performance under three distinct baseline augmentation strategies: data generation, token replacement, and padding enhancement. The specific baselines are introduced as follows:

Figure 2: Illustration of AUC, NDCG@10, and MRR Performance across Entropy Thresholds.

- **HSTU** Zhai et al. (2024) introduces a generative sequential transduction architecture specialized for high-cardinality, non-stationary streaming recommendation tasks.
- **LLaMA2Rec** Touvron et al. (2023) adapts the LLaMA2 transformer architecture to sequential recommendation and fine-tunes it for domain-specific tasks.
- **RepPad** Dang et al. (2024a) introduces a sequence-level data augmentation strategy for recommendation by replacing zero-padding with repeated interactions to optimize input utilization.
- **FRec** Li et al. (2024) introduces a fatigue-aware sequence augmentation technique, embedding explicit user fatigue signals to enhance contrastive learning in sequential recommendation.
- **DR4SR** Yin et al. (2024) introduces a data-centric dataset regeneration framework for sequential recommendation, enabling model-agnostic and personalized data regeneration.

We divide the experimental validation into two parts: Overall Performance and Further Validation. In Overall Performance, we provide a detailed comparison of UniDS with various baselines under different backbone architectures. In Further Validation, we separately verify the Metric-oriented Gradient Comparison Theory in Section 5.2.1, Pattern Mining via Conditional Entropy in Section 5.2.2, and Uniform Dataset Generation in Section 5.2.3.

**Experiment Settings.** We adopt the leave-one-out strategy for evaluation in the recall and ranking stages, following prior researchKang & McAuley (2018b); Sun et al. (2019b). Specifically, for each user sequence, the most recent interaction is reserved for testing, the second most recent for validation, and the remainder for training. In the recall stage, we evaluate Top-K recommendation performance using MRRCraswell (2016) and HRWaters (1976). In the coarse ranking (rough ranking) stage, we use NDCGJärvelin & Kekäläinen (2002) for evaluation, while in the fine ranking (re-ranking) stage, AUC Bradley (1997) and Logloss are adopted to assess model performance. Details regarding equipment, training time, and other hyperparameter settings are provided in Appendix A.2.

## 5.1 OVERALL PERFORMANCE

In this subsection, we provide a comprehensive comparison between UniDS and all competitive baselines, as summarized in Table 1. Our key findings are as follows: 1). UniDS consistently achieves the best overall results across all datasets and evaluation tasks, demonstrating significant improvements over both traditional and generative baselines. This highlights UniDS's strong generalization ability and validates its design for robust multi-stage recommendation. 2) The adoption of entropy-driven pattern segmentation in UniDS enables unified data quality control throughout the recommendation pipeline, leading to substantial performance gains over methods such as RepPad and FRec. This result underscores the importance of multi-stage quality alignment for maximizing end-to-end effectiveness. 3). By integrating semantic token representation with entropy-adaptive behavioral pattern modeling in a unified generative framework, UniDS further advances beyond approaches like DR4SR. This cohesive generation paradigm facilitates stable and broadly applicable improvements, especially in challenging domains with varying data sparsity. 4) These findings confirm that entropy-based dataset unification plays a pivotal role in achieving interpretable, resilient, and scalable enhancements for multi-stage recommendation systems, advancing the robustness of sequential models in both dense and sparse data scenarios.

## 5.2 FURTHER VALIDATION EXPERIMENTS

### 5.2.1 EXPERIMENT VALIDATION ON THEOREM 4.1

We systematically examine the influence of increasing sequence length (data quantity) and applying data generation (data quality enhancement) on recommendation performance, measured by MRR, NDCG@10, and AUC to respectively reflect the recall, coarse-ranking, and fine-ranking stages of

Table 1: Results comparison between UniDS and data augmentation baselines are conducted under two frameworks: HSTU and LLaMA2Rec, where FRT denotes the Fine Ranking Task. * indicates statistically significant differences with p < 0.05.

| Baselines | Different Models | HSTU | | | | | | LLaMA2Rec | | | | | |
|---|---|---|---|---|---|---|---|---|---|---|---|---|---|
| | | AUC FRT | MRR | HR@50 Recall Task | HR@10 | NG@50 Coarse Ranking Task | NG@10 | AUC FRT | MRR | HR@50 Recall Task | HR@10 | NG@50 Coarse Ranking Task | NG@10 |
| ML-1M | Base | 0.7861 | 0.1535 | 0.5913 | 0.3271 | 0.2409 | 0.1824 | 0.7351 | 0.1320 | 0.5472 | 0.2806 | 0.2135 | 0.1547 |
| | DR4SR | 0.7931 | 0.1540 | 0.5925 | 0.3351 | 0.2434 | 0.1846 | 0.7401 | 0.1366 | 0.5549 | 0.2938 | 0.2182 | 0.1601 |
| | RepPad | 0.7903 | 0.1559 | 0.5923 | 0.3263 | 0.2430 | 0.1838 | 0.7381 | 0.1297 | 0.5252 | 0.2705 | 0.2065 | 0.1510 |
| | FRec | 0.7945 | 0.1550 | 0.5905 | 0.3249 | 0.2420 | 0.1829 | 0.7412 | 0.1316 | 0.5301 | 0.2729 | 0.2095 | 0.1527 |
| | UniDS | 0.8081* | 0.1598* | 0.6044* | 0.3372* | 0.2447* | 0.1908* | 0.7485* | 0.1435* | 0.5615* | 0.2957* | 0.2250* | 0.1675* |
| KR-Pure | Base | 0.7351 | 0.0542 | 0.3158 | 0.1150 | 0.1019 | 0.0586 | 0.7251 | 0.0519 | 0.2975 | 0.1103 | 0.0971 | 0.0568 |
| | DR4SR | 0.7401 | 0.0624 | 0.3346 | 0.1267 | 0.1106 | 0.0658 | 0.7370 | 0.0572 | 0.3231 | 0.1209 | 0.1059 | 0.0623 |
| | RepPad | 0.7381 | 0.0567 | 0.3264 | 0.1256 | 0.1049 | 0.0609 | 0.7367 | 0.0565 | 0.3208 | 0.1210 | 0.1047 | 0.0614 |
| | FRec | 0.7412 | 0.0578 | 0.3336 | 0.1236 | 0.1062 | 0.0639 | 0.7355 | 0.0568 | 0.3202 | 0.1191 | 0.1049 | 0.0615 |
| | UniDS | 0.7492* | 0.0668* | 0.3485* | 0.1346* | 0.1146* | 0.0689* | 0.7427* | 0.0592* | 0.3371* | 0.1239* | 0.1102* | 0.0640* |
| AMZ-Books | Base | 0.7519 | 0.0249 | 0.1159 | 0.0502 | 0.0416 | 0.0273 | 0.7490 | 0.0270 | 0.1163 | 0.0533 | 0.0435 | 0.0298 |
| | DR4SR | 0.7530 | 0.0274 | 0.1260 | 0.0560 | 0.0456 | 0.0304 | 0.7521 | 0.0279 | 0.1212 | 0.0554 | 0.0452 | 0.0309 |
| | RepPad | 0.7526 | 0.0294 | 0.1346 | 0.0594 | 0.0488 | 0.0325 | 0.7516 | 0.0275 | 0.1186 | 0.0537 | 0.0442 | 0.0302 |
| | FRec | 0.7540 | 0.0279 | 0.1212 | 0.0554 | 0.0452 | 0.0309 | 0.7427 | 0.0278 | 0.1200 | 0.0545 | 0.0447 | 0.0305 |
| | UniDS | 0.7620* | 0.0323* | 0.1351* | 0.0629* | 0.0514* | 0.0357* | 0.7605* | 0.0302* | 0.1226* | 0.0580* | 0.0473* | 0.0333* |

Table 2: Performance improvement rates on different sequence lengths with and without generated data. "Gen vs. Origin (%)" reports relative improvements for each metric: mean reciprocal rank (MRR), normalized discounted cumulative gain at 10 (NDCG@10), and area under the ROC curve (AUC). "Longer Seql Adv. (%)" reports the relative improvement (%) on each metric when increasing MAX_Seqlen, calculated as $(CurrentSeqLen - ShorterSeqLen)/ShorterSeqLen \times 100\%$.

| Dataset | MAX_Seqlen | Method | Tokens | MRR | NDCG@10 | AUC | Gen vs. Origin (%) | Longer Seql Adv. (%) |
|---|---|---|---|---|---|---|---|---|
| ML-1M | 100 | Origin | 505,108 | 0.1482 | 0.1780 | 0.7790 | – | – |
| | | Generated | 859,700 | 0.1549 | 0.1820 | 0.7940 | 4.52 / 2.25 / 1.93 | – |
| | 150 | Origin | 802,493 | 0.1532 | 0.1830 | 0.7820 | – | 3.37 / 2.81 / 0.39 |
| | | Generated | 1,105,884 | 0.1563 | 0.1872 | 0.8020 | **2.02 / 2.30 / 2.56** | 0.90 / 2.86 / 1.01 |
| | 200 | Origin | 992,827 | 0.1535 | 0.1840 | 0.7865 | – | **0.20 / 0.55 / 0.58** |
| | | Generated | 1,286,790 | 0.1565 | 0.1878 | 0.8080 | **1.95 / 2.07 / 2.73** | 0.13 / 0.32 / 0.75 |
| KR-Pure | 25 | Origin | 560,136 | 0.0504 | 0.0609 | 0.7221 | – | – |
| | | Generated | 566,001 | 0.0530 | 0.0601 | 0.7301 | 3.37 / 2.30 / 1.11 | – |
| | 50 | Origin | 911,752 | 0.0521 | 0.0623 | 0.7301 | – | 6.55 / 5.09 / 0.18 |
| | | Generated | 914,690 | 0.0542 | 0.0648 | 0.7354 | **0.93 / 1.25 / 1.66** | 4.03 / 4.01 / 0.73 |
| | 100 | Origin | 1,245,535 | 0.0542 | 0.0647 | 0.7351 | – | **0.93 / 1.09 / 1.62** |
| | | Generated | 1,251,400 | 0.0550 | 0.0657 | 0.7485 | **1.48 / 1.55 / 1.82** | 1.48 / 1.39 / 1.78 |

**Note:** "Gen vs. Origin (%)" columns show the improvement rates of MRR, NDCG@10, and AUC (in order) for generated data compared to original data: $(Generated - Origin)/Origin \times 100\%$. "Longer Seql Adv. (%)" columns show the improvement for increased sequence length ("% MRR / NDCG@10 / AUC", in order): $(CurrentSeqLen - ShorterSeqLen)/ShorterSeqLen \times 100\%$.

the model. Key findings from Table 2 are as follows: 1). Extending sequence length universally improves all evaluation metrics, with the largest relative gains observed in shorter sequences. This demonstrates that increasing data quantity primarily benefits the early recall and coarse-ranking stages, with MRR and NDCG@10 showing more pronounced improvements when sequence length is small. 2). Data generation yields additional performance improvements, especially as sequence length increases. Notably, the enhancement is most significant for the fine-ranking stage (AUC) on long sequences. This pattern indicates that the effect of data quality amplification grows stronger when behavioral information is abundant, supporting the ordering: $\frac{\delta AUC}{\delta S^{real}} \geq \frac{\delta NDCG}{\delta S^{real}} \geq \frac{\delta MRR}{\delta S^{real}}$, as proved in Theorem 4.1. In other words, as the real entropy of the user sequence increases, the sensitivity and benefit for fine-ranking (AUC) surpass those for coarse-ranking (NDCG@10) and recall (MRR).

### 5.2.2 Validation on Pattern Mining via Conditional Entropy

We investigate how varying the global entropy threshold $\theta$ in our pattern mining framework impacts downstream performance, which is illustrated in Fig. 2. From Fig. 2, we can draw the following conclusions: 1) Increasing the threshold allows the miner to extract longer and more complex low-entropy patterns from user sequences. This selectively amplifies data quality and yields pronounced improvements in fine-ranking metrics (AUC), especially as behavioral information becomes more abundant. Such patterns enhance the model's ability to capture nuanced sequence dependencies, directly contributing to improved fine-ranking accuracy. 2) Meanwhile, coarse-ranking (NDCG@10) and recall (MRR) metrics respond differently: their performance peaks at lower thresholds, indicating that these tasks benefit most from the extraction of shorter, simpler patterns. This reduced sensitivity

Table 3: Ablation study of UniDS, where w/o TC denotes removing threshold control, w/o UF denotes removing Uniform Data Formalization, and w/o UG denotes removing Uniform Data Generation.

| Method | ML-1M | | | | KR-Pure | | | | AMZ-Books | | | |
|---|---|---|---|---|---|---|---|---|---|---|---|---|
| | AUC | NG10 | HR10 | MRR | AUC | NG10 | HR10 | MRR | AUC | NG10 | HR10 | MRR |
| w/o TC | 0.8030 | 0.1805 | 0.3179 | 0.1538 | 0.7455 | 0.0677 | 0.1322 | 0.0616 | 0.7570 | 0.0317 | 0.0569 | 0.0288 |
| w/o UF | 0.7928 | 0.1842 | 0.3345 | 0.1538 | 0.7399 | 0.0655 | 0.1264 | 0.0622 | 0.7528 | 0.0301 | 0.0557 | 0.0272 |
| w/o UG | 0.7863 | 0.1837 | 0.3268 | 0.1534 | 0.7350 | 0.0583 | 0.1146 | 0.0539 | 0.7515 | 0.0271 | 0.0500 | 0.0247 |
| UniDS | **0.8081** | **0.1878** | **0.3322** | **0.1565** | **0.7494** | **0.0698** | **0.1359** | **0.0630** | **0.7620** | **0.0357** | **0.0629** | **0.0323** |

to data quality amplification suggests that recall-based and coarse-ranking stages rely more on salient but less complex behavioral subsequences. Importantly, these experimental observations illustrate the effectiveness of our architecture's adjustable entropy threshold $\theta$, which unifies and accommodates the different data quality demands at separate ranking stages. By tuning entropy threshold $\theta$, our framework flexibly generates pattern-based datasets that optimize performance for both fine- and coarse-grained recommendation objectives. Overall, these results confirm that conditional entropy pattern mining not only segments user behavior into robust, interpretable subsequences but also provides a unified and customizable data generation process for diverse downstream tasks in SR Systems.

### 5.2.3 ABLASION ON UNIFORM DATASET GENERATION

To investigate the impact of Uniform Data Generation (UG) on model performance, we conduct an ablation study by incrementally removing key components from the UniDS framework. Table 3 reports results across three datasets: ML-1m, Krnd, and Books. Specifically, we evaluate variants that exclude threshold control (w/o TC), Uniform Data Formalization (w/o UF), and Uniform Data Generation (w/o UG). The ablation results yield two main findings: 1) The exclusion of UG leads to a consistent and substantial reduction in all metrics across all datasets. This demonstrates that UG is essential for alleviating data imbalance and improving model generalization, particularly in scenarios with highly skewed or sparse data distributions. The performance drop after removing UG underscores its role in reducing bias and supporting more accurate recommendations. 2) The complete UniDS framework achieves the highest scores on all metrics, outperforming variants with individual components removed. This indicates that UG, when combined with threshold control and data formalization, provides complementary benefits that reinforce each other. The unified integration of these mechanisms is crucial for realizing the full potential of the model, as opposed to relying on isolated improvements from any single component.

## 6 DISCUSSION

**Broader Impact and Future Directions**. Many components of our theoretical framework and its applications are highly flexible. For instance, when trying to adapt UniSR into LLM Agent tasks, both Real Entropy ($S^{real}$) and User Predictability Rate ($\Pi_{max}$) can be preserved, as they are derived from discrete sequences. In this context, user sequences from multi-stage SR can be directly replaced by textual sequences generated by the LLM Agent. Similarly, the pattern miner within our unified data quality framework is capable of segmenting textual sequences, and the inclusion of a Special Token in our unified data format designates specific positions for tool-calling tokens required at various stages of Agent tasks. In future work, we plan to further extend UniDS to encompass a broader spectrum of multi-stage tasks.

**Conclusion**. In summary, this work introduces UniDS, a pioneering unified framework for data synthesis in multi-stage complex structures, effectively bridging the gap between existing coarse multi-stage approaches. Through the development of Metric-oriented Gradient Comparison Theory and a conditional entropy-based pattern mining module, our framework offers a theoretically grounded Real Entropy metric and adaptive pattern extraction capabilities, facilitating unified and adaptive entropy control. The Uniform Dataset Generation paradigm, built upon standardized Pattern-Token sequence representations and unified generation logic, ensures flexible and consistent feature synthesis across various stages. Both rigorous theoretical analysis and extensive empirical evaluations on benchmark datasets validate the effectiveness of UniDS, demonstrating notable improvements in recommendation performance, enhanced flexibility in feature synthesis, and robust stage adaptation. Collectively, these contributions establish UniDS as an effective, model-agnostic foundation for high-quality data generation in complex multi-stage applications.

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

Table 4: The basic information for different datasets.

| Dataset Details, $\#Tokens = \Sigma_u |S_u|$ | | | |
|---|---|---|---|
| Dataset | | $|S_u|_{max}$ | $\#Tokens$ |
| KR-Pure | User: 27,285 Item: 7,551 | 25 | 560,136 |
| | | 50 | 911,752 |
| | | 100 | 1,245,535 |
| ML-1M | User: 6,040 Item: 3,706 | 100 | 505,108 |
| | | 150 | 802,493 |
| | | 200 | 1,058,511 |
| AMZ-Books | User: 694,897 Item: 686,623 | 50 | 8,044,865 |
| | | 25 | 7,076,238 |

Guorui Zhou, Jiaxin Deng, Jinghao Zhang, Kuo Cai, Lejian Ren, Qiang Luo, Qianqian Wang, Qigen Hu, Rui Huang, Shiyao Wang, Weifeng Ding, Wuchao Li, Xinchen Luo, Xingmei Wang, Zexuan Cheng, Zixing Zhang, Bin Zhang, Boxuan Wang, Chaoyi Ma, Chengru Song, Chenhui Wang, Di Wang, Dongxue Meng, Fan Yang, Fangyu Zhang, Feng Jiang, Fuxing Zhang, Gang Wang, Guowang Zhang, Han Li, Hengrui Hu, Hezheng Lin, Hongtao Cheng, Hongyang Cao, Huanjie Wang, Jiaming Huang, Jiapeng Chen, Jiaqiang Liu, Jinghui Jia, Kun Gai, Lantao Hu, Liang Zeng, Liao Yu, Qiang Wang, Qidong Zhou, Shengzhe Wang, Shihui He, Shuang Yang, Shujie Yang, Sui Huang, Tao Wu, Tiantian He, Tingting Gao, Wei Yuan, Xiao Liang, Xiaoxiao Xu, Xugang Liu, Yan Wang, Yi Wang, Yiwu Liu, Yue Song, Yufei Zhang, Yunfan Wu, Yunfeng Zhao, and Zhanyu Liu. Onerec technical report, 2025. URL https://arxiv.org/abs/2506.13695.

Kun Zhou, Hui Wang, Wayne Xin Zhao, Yutao Zhu, Sirui Wang, Fuzheng Zhang, Zhongyuan Wang, and Ji-Rong Wen. S3-rec: Self-supervised learning for sequential recommendation with mutual information maximization. In Mathieu d'Aquin, Stefan Dietze, Claudia Hauff, Edward Curry, and Philippe Cudré-Mauroux (eds.), *CIKM '20: The 29th ACM International Conference on Information and Knowledge Management, Virtual Event, Ireland, October 19-23, 2020*, pp. 1893–1902. ACM, 2020. doi: 10.1145/3340531.3411954. URL https://doi.org/10.1145/3340531.3411954.

Zhuang Zhuang, Tianxin Wei, Lingbo Liu, Heng Qi, Yanming Shen, and Baocai Yin. Tau: Trajectory data augmentation with uncertainty for next poi recommendation. In *AAAI*, 2024.

Jacob Ziv and Abraham Lempel. A universal algorithm for sequential data compression. *IEEE Transactions on information theory*, 23(3):337–343, 1977.

## A APPENDIX / SUPPLEMENTAL MATERIAL

### A.1 DETAILS ON DATASET

We present the specific details of the dataset in Table 4.

### A.2 DETAILED EXPERIMENT SETTINGS

Following previous works Kang & McAuley (2018b); Sun et al. (2019b); Zhou et al. (2020), we leverage the leave-one-out method to calculate the recommendation performance. Besides, we adopt the whole item set as the candidate item set during evaluation to avoid the sampling bias of the candidate selection Krichene & Rendle (2020). Then, we evaluate the Top-K recommendation performance by Normalized Discounted Cumulative Gain (NDCG) Järvelin & Kekäläinen (2002) and Hit Rate (HR) Waters (1976). To effectively demonstrate the performance of models with varying parameters across different datasets, we selected different parameters for fitting based on the size of each dataset. Regarding model configurations, for the MovieLens-1M, KuaiRand-pure, and Amazon Books datasets, we configured $N \in \{4, 8, 12, 16, 24, 32\}$ and $d_{emb} \in \{25, 50, 75, 100\}$. From a data perspective, we selected the maximum sequence length for truncation based on the average length of each dataset. In the MovieLens-1M dataset, we selected according to the maximum sequence length $S_{max} \in \{100, 150, 200\}$. In the KuaiRand-Pure dataset, we set the maximum sequence length $S_{max} \in \{25, 50, 100\}$. Finally, for the Amazon Books, we configured the maximum sequence

length $S_{max} \in \{25, 50\}$. The detail parameter are: hidden_size=128, layer_num=2, head_num=2, dropout_rate=0.5, activation='gelu', and layer_norm_eps=1e-12. All other parameters are set to the default values as reported for HSTU in the original paper and public implementations. The largest model we executed reached a model depth of $N = 32$, an embedding dimension of $d_{emb} = 100$, and a vocabulary size of $|I| = 686, 623$. We utilized 8 industrial GPUs to run this experiment, with the largest experiment taking 4 hours. This truly allowed us to study model performance at extreme data and model scales.

### A.3 PROOF OF THEOREM 4.1

**Theorem A.1.** *Assuming $rank_i$ denotes the rank assigned by the model to the $i$-th target positive item, $\overline{rank_i}$ represents the model-assigned rank for the item to be predicted at the top position, and $S^{real}$ denotes the real entropy of the user sequence, the sensitivity of the fine-ranking metric AUC, the coarse-ranking metric NDCG, and the recall metric MRR to real entropy is ordered as follows:*

$$\frac{\delta AUC}{\delta S^{real}} = -o(\frac{1}{rank_i}) \geq \frac{\delta NDCG}{\delta S^{real}} = -o(\frac{1}{rank_i \log_2(\overline{rank_i})^2}) \geq \frac{\delta MRR}{\delta S^{real}} = -o(\frac{1}{rank_i^2}) \quad (9)$$

*Proof.* We bridge the evaluation metrics and the real entropy $S^{real}$ of user interaction sequences through the model's predictability $\Pi_{max}$ and the predicted item ranking $rank_i$. Here, $rank_i$ denotes the position assigned by the model to the $i$-th positive instance, and $\Pi_{max}$ is the maximum probability with which the model correctly predicts the next target item. Let $I$ denote the whole item set, $I^+$ the set of ground-truth positives, $N_+$ the number of positives, $N_-$ the number of negatives in the candidate pool, and $M = N_+ + N_-$ the candidate set size. $\overline{rank_i}$ denotes the model-assigned rank for the top-predicted positive item.

For AUC, which estimates the probability that a randomly chosen positive is ranked above a randomly chosen negative, the standard formulation is:

$$AUC = \frac{\Sigma_{i \in I^+}(N_+ + N_- - rank_i) - \frac{M(M+1)}{2}}{N_+ N_-} \quad (10)$$

Applying the chain rule, the sensitivity of AUC to the entropy $S^{real}$ can be written as:

$$\frac{\delta AUC}{\delta S^{real}} = \frac{\Sigma_{i \in I^+}(N_+ + N_- - rank_i) - \frac{M(M+1)}{2}}{N_+ N_-} \frac{1}{\delta S^{real}} \quad (11)$$

Rearranging the terms yields:

$$\frac{\delta AUC}{\delta S^{real}} = \delta \Sigma_{i \in I^+} \delta \frac{\Sigma_{i \in I^+}(N_+ + N_- - rank_i)}{N_+ N_-} \Sigma_{i \in I^+} \frac{1}{\delta S^{real}} \quad (12)$$

Noting that AUC depends linearly on $rank_i$, we can simplify the above as:

$$\frac{\delta AUC}{\delta S^{real}} = -\frac{\Sigma_{i \in I^+} \delta \, rank_i}{N_+ N_-} \frac{1}{\delta S^{real}} \quad (13)$$

Next, the real entropy $S^{real}$, relating to model predictability $\Pi_{max}$, is given by:

$$S^{real} = -(\Pi_{max} \log_2 \Pi_{max} + (1 - \Pi_{max}) \log_2(1 - \Pi_{max})) + (1 - \Pi_{max}) \log_2(|I| - 1) \quad (14)$$

The derivative of $S^{real}$ with respect to $\Pi_{max}$ is:

$$\delta S^{real} = -(\log_2 \Pi_{max} + \frac{\Pi_{max}}{\Pi_{max} \ln 2} - \log_2(1 - \Pi_{max}) +$$
$$(1 - \Pi_{max})(-\frac{1}{(1 - \Pi_{max}) \ln 2})) \delta \Pi_{max} - \log_2(|I| - 1) \delta \Pi_{max} \quad (15)$$

This can be further simplified to:

$$\delta S^{real} = -(\frac{\ln(\Pi_{max}) + 1}{\ln 2} - \frac{\ln(1 - \Pi_{max}) + 1}{\ln 2}) \delta \Pi_{max} - \log_2(|I| - 1) \delta \Pi_{max} \quad (16)$$

$$\delta S^{real} = \log_2 \frac{(1 - \Pi_{max})}{\Pi_{max}(N-1)} \delta \Pi_{max} \tag{17}$$

Substituting this result, the change of AUC with respect to $S^{real}$ is:

$$\frac{\delta AUC}{\delta S^{real}} = -\frac{\Sigma_{i \in I^+} \delta \, rank_i}{N_+ N_-} \cdot \frac{1}{\log_2 \frac{(1-\Pi_{max})}{\Pi_{max}(N-1)} \delta \Pi_{max}} \tag{18}$$

$$\frac{\delta AUC}{\delta S^{real}} = -\frac{1}{N_+ N_- \log_2 \frac{(1-\Pi_{max})}{\Pi_{max}(N-1)}} \frac{\Sigma_{i \in I^+} \delta \, rank_i}{\delta \Pi_{max}} \tag{19}$$

$$\frac{\delta AUC}{\delta S^{real}} = -\frac{1}{N_- \log_2 \frac{(1-\Pi_{max})}{\Pi_{max}(N-1)}} \frac{\Sigma_{i \in I^+} \overline{\delta \, rank_i}}{\delta \Pi_{max}} \tag{20}$$

Given that the expected change of rank for positive items is typically inversely proportional to their predicted rank, we ultimately get:

$$\frac{\delta AUC}{\delta S^{real}} = -o\left(\frac{1}{rank_i}\right) \frac{1}{\log_2 \frac{(1-\Pi_{max})}{\Pi_{max}(N-1)}} \frac{\Sigma_{i \in I^+} \overline{\delta \, rank_i}}{\delta \Pi_{max}} \tag{21}$$

For the coarse-ranking metric NDCG (Normalized Discounted Cumulative Gain), defined as

$$NDCG = \sum_{i \in I^+} \frac{1}{\log_2(\text{rank}_i + 1)} \tag{22}$$

the variation in NDCG with respect to the predicted ranks can be written as

$$\delta NDCG = -\sum_{i \in I^+} \frac{\ln 2 \cdot \delta \, \text{rank}_i}{(\text{rank}_i + 1) \, [\log_2(\text{rank}_i + 1)]^2} \tag{23}$$

where we use the derivative of the log term with respect to rank.

Thus, the sensitivity of NDCG to the change in real entropy yields:

$$\frac{\delta NDCG}{\delta S^{real}} = -\sum_{i \in I^+} \frac{\ln 2 \cdot \delta \, \text{rank}_i}{(\text{rank}_i + 1) \, [\log_2(\text{rank}_i + 1)]^2} \frac{1}{\delta S^{real}} \tag{24}$$

where $\delta S^{real}$ is as previously derived. Substituting this in, we have

$$\frac{\delta NDCG}{\delta S^{real}} = -\sum_{i \in I^+} \frac{\ln 2 \cdot \delta \, \text{rank}_i}{(\text{rank}_i + 1) \, [\log_2(\text{rank}_i + 1)]^2} \frac{1}{\log_2\left(\frac{1-\Pi_{max}}{\Pi_{max}(N-1)}\right) \delta \Pi_{max}} \tag{25}$$

Focusing only on the dominant term (i.e., only considering the top-ranked positive item), we approximate:

$$\frac{\delta NDCG}{\delta S^{real}} = -\frac{\ln 2 \cdot \overline{\delta \, \text{rank}_i}}{(\overline{\text{rank}_i} + 1) \, [\log_2(\overline{\text{rank}_i} + 1)]^2} \cdot \frac{1}{\log_2\left(\frac{1-\Pi_{max}}{\Pi_{max}(N-1)}\right) \delta \Pi_{max}} \tag{26}$$

By gathering the dominant asymptotic behavior, we have

$$\frac{\delta NDCG}{\delta S^{real}} = -o\left(\frac{1}{\overline{\text{rank}_i} \cdot [\log_2(\overline{\text{rank}_i})]^2}\right) \cdot \frac{1}{\log_2\left(\frac{1-\Pi_{max}}{\Pi_{max}(N-1)}\right)} \frac{\sum_{i \in I^+} \overline{\delta \, \text{rank}_i}}{\delta \Pi_{max}} \tag{27}$$

For the recall-style metric MRR (Mean Reciprocal Rank):

$$MRR = \sum_{i \in I^+} \frac{1}{\text{rank}_i} \tag{28}$$

The differential is

$$\delta MRR = -\sum_{i \in I^+} \frac{1}{\text{rank}_i^2} \delta \, \text{rank}_i \tag{29}$$

Thus, the sensitivity to entropy $S^{real}$ is

$$\frac{\delta MRR}{\delta S^{real}} = -\sum_{i \in I^+} \frac{1}{\text{rank}_i^2} \delta \text{rank}_i \cdot \frac{1}{\delta S^{real}} \tag{30}$$

$$\frac{\delta MRR}{\delta S^{real}} = -\sum_{i \in I^+} \frac{\delta \text{rank}_i}{\text{rank}_i^2} \cdot \frac{1}{\log_2\left(\frac{1-\Pi_{max}}{\Pi_{max}(N-1)}\right) \delta \Pi_{max}} \tag{31}$$

Again, focusing on the top-ranked item only, we obtain

$$\frac{\delta MRR}{\delta S^{real}} = -o\left(\frac{1}{\text{rank}_i^{-2}}\right) \cdot \frac{1}{\log_2\left(\frac{1-\Pi_{max}}{\Pi_{max}(N-1)}\right)} \frac{\sum_{i \in I^+} \overline{\delta \text{rank}_i}}{\delta \Pi_{max}} \tag{32}$$

Collectively, these results clearly demonstrate the order of sensitivity for AUC, NDCG and MRR to the real entropy $S^{real}$:

$$\frac{\delta AUC}{\delta S^{real}} = -o\left(\frac{1}{\text{rank}_i}\right) \geq \frac{\delta NDCG}{\delta S^{real}} = -o\left(\frac{1}{\text{rank}_i[\log_2(\text{rank}_i)]^2}\right) \geq \frac{\delta MRR}{\delta S^{real}} = -o\left(\frac{1}{\text{rank}_i^2}\right) \tag{33}$$

where all omitted constant and higher-order terms are identical (dominated by the denominator factor $\log_2\left(\frac{1-\Pi_{max}}{\Pi_{max}(N-1)}\right)$), and $\overline{\text{rank}_i}$ denotes the rank assigned to the top-predicted positive item. This completes the proof. $\square$

### A.4 LLM Usage

Large language models were employed in this study as general assistive tools to improve the overall clarity and readability of this paper. Specifically, these models were used to refine the grammar, punctuation, and phrasing within the manuscript. No language models were utilized to generate original scientific ideas, perform data analysis, or derive scientific conclusions. All scientific content, including experimental design, data analysis, and interpretation, was solely conducted by the authors.

### A.5 Dynamic Programming Algorithm for Pattern Mining via Conditional Entropy

### A.6 Details on Uniform Data Generation Model

#### A.6.1 Construction of Pre-training Task

The objective of our pre-training task is to train the dataset regenerator $R_\phi$ to effectively reconstruct high-value item transition patterns identified from the original dataset $\mathcal{D}$ in a self-supervised manner. To achieve this, we first mine prevalent transition patterns $\mathcal{P}$ using a rule-based sliding window approach.

Specifically, given a user sequence $s_u = [i_1, i_2, \ldots, i_L]$, we extract all contiguous subsequences of length $\alpha$ via a sliding window:

$$w_{u,k} = [i_k, i_{k+1}, \ldots, i_{k+\alpha-1}], \quad k = 1, 2, \ldots, L - \alpha + 1$$

For each window $w_{u,k}$, we regard the ordered tuple as a candidate transition pattern. The frequency of each pattern $\pi$ across all user sequences is:

$$f(\pi) = \sum_u \sum_k \mathbb{I}[w_{u,k} = \pi]$$

Patterns whose frequency exceeds a minimum threshold $\beta$ are selected:

$$\mathcal{P} = \{\pi \mid f(\pi) \geq \beta\}$$

---

**Algorithm 1:** Entropy-driven Pattern Extraction via Conditional Entropy and Support

---

**Input** : User sequences $\{S_u\}$; model $p_m$; thresholds $\theta, c, L_{\max}$
**Output**: Set of robust, entropy-driven patterns $\mathcal{E}$

Compute empirical frequencies $f_i$ and total $F$;
Calculate initial item entropies $H(i)$ using model $p_m$;
Set global threshold $\theta$ (scaled mean entropy);

**Initialize seeds:**;
$\mathcal{P} \leftarrow \{[i] : H(i) < \theta, \text{ Occu}([i]) \geq c\}$

**Iterative extension:**;
**while** $\mathcal{P}$ *is not empty* **do**
  $\mathcal{P}_{new} \leftarrow \emptyset$;
  **foreach** $P = [i_1, \ldots, i_n] \in \mathcal{P}$ **do**
    **foreach** *observed successor $s$ after $P$ in user data* **do**
      $P' = [i_1, \ldots, i_n, s]$;
      Compute incremental entropy $h(s \mid P)$ using $p_m$;
      $H(P') = H(P) + h(s \mid P)$; // DP caching
      **if** $H(P') < \theta$ **and** $|P'| \leq L_{\max}$ **and** $Occu(P') \geq c$ **then**
        Add $P'$ to $\mathcal{P}_{new}$;
        Add $P'$ to $\mathcal{E}$;

  Replace $\mathcal{P} \leftarrow \mathcal{P}_{new}$;

**return** $\mathcal{E}$

---

**Autoregressive Modeling and Objective**  To leverage these mined patterns for self-supervised pre-training, we cast the regenerator $R_\phi$ as an autoregressive sequence model. For a given sequence $s_u = [i_1, i_2, \ldots, i_L]$, $R_\phi$ is trained to estimate the conditional probability of the next item given the preceding context:

$$P_\phi(i_t \mid i_{1:t-1}), \quad t = 2, 3, \ldots, L$$

For each pattern $\pi = [j_1, j_2, \ldots, j_\alpha] \in \mathcal{P}$, we enforce that $R_\phi$ can reconstruct $\pi$ by maximizing the likelihood of observed transitions. Specifically, for each pattern occurrence in $\mathcal{D}$, the model is optimized with the following autoregressive loss:

$$\mathcal{L}_{\text{AR}} = -\sum_{\pi \in \mathcal{P}} \sum_{t=1}^{\alpha} \log P_\phi(j_t \mid j_{1:t-1})$$

where $j_{1:0}$ is defined as the special start-of-sequence token.

Aggregating over all occurrences of mined patterns, the overall pre-training objective becomes:

$$\mathcal{L}_{\text{pretrain}} = -\sum_{u} \sum_{k} \sum_{t=1}^{\alpha} \mathbb{I}[w_{u,k} \in \mathcal{P}] \cdot \log P_\phi(i_{k+t-1} \mid i_{k:k+t-2})$$

That is, $R_\phi$ learns to maximize the likelihood of generating each item in high-value transition patterns, conditioned on prior item history within each mined sliding window.

### A.6.2  DIVERSITY-PROMOTED REGENERATOR

With the aid of the pre-training task, we can now pre-train a dataset regenerator. In this paper, we employ the Transformer model as the main architecture of our regenerator, whose generation capabilities have been extensively validated. The dataset regenerator consists of three modules: an encoder to obtain representations of sequences in the original dataset, a decoder to regenerate the patterns, and a diversity promoter to capture the one-to-many mapping relationship. Next, we will proceed to introduce each of these modules individually.

The encoder consists of several stacked layers of multi-head self-attention (MHSA) and feed-forward (FFN) layers. Considering a sequence in $\mathcal{X}$, we can get the sequence representation by:

$$\mathbf{H}^{(l)} = FFN(MHSA(\mathbf{H}^{(l-1)})), \tag{34}$$

where $\mathbf{H}^{(l-1)} \in \mathbb{R}^{N \times d}$ is the output at the $(l-1)$-th layer, and $\mathbf{H}^{(0)}$ is item embeddings with added learnable positional encoding. As for the decoder, it takes patterns in the regenerated dataset $\mathcal{X}'$ as input. The objective of the decoder is to reconstruct the pattern given the sequence representations from the encoder.

In our approach, the input sequence and pattern $(s_u, p_i)$ are transformed into the item-behavior-item-behavior format: $s_u$ is represented as $[i_1, b_1, i_2, b_2, \ldots]$ and similarly, $p_i$ is represented as $[p_{i1}, b_{i1}, p_{i2}, b_{i2}, \ldots]$.

$$L_{\text{recon}} = -\sum_{(s_u, p_i)}^{|\mathcal{X}_{\text{pre}}|} \sum_{t=1}^{T} \log P(p_{it}, b_{it} | \mathbf{h}_u^{(l)}, \hat{p}_{<t}), \tag{35}$$

where $(s_u, p_i)$ is a sequence-pattern pair in $\mathcal{X}_{\text{pre}}$, $t$ is the position in the target pattern, $\mathbf{h}_u^{(l)}$ is the representation of the original sequence formatted as item-behavior-item-behavior , $\hat{p}_{<t}$ is the pattern generated before $t$, and $P$ is the prediction probability for the target item $p_{it}$.

However, as mentioned in Section A.6.1, multiple patterns can be extracted from a sequence, which presents challenges during the training process. For instance, a sequence like (1, 2, 3, 4, 5) may yield two distinct patterns, namely (1, 2) and (4, 5). Consequently, this can introduce conflicts during training, thereby impeding the convergence of the dataset regenerator. To address this one-to-many mapping issue, we further propose a diversity promoter.

Specifically, we employ a more aggressive approach to adaptively regulate the impact of the original sequence during the decoding stage by incorporating information about the target pattern. First, we project the memory $\mathbf{m} \in \mathbb{R}^D$ (representations of the original sequence) generated by the encoder into K distinct vector spaces, i.e., $\{\mathbf{m}_1', \mathbf{m}_2', \ldots, \mathbf{m}_K'\}$ and $\mathbf{m}_k' \in \mathbb{R}^D$. This projection enables the acquisition of memories with diverse semantic information. Ideally, different target patterns should match distinct memories. To achieve this, we additionally introduce a Transformer encoder to encode the target patterns and obtain $\mathbf{h}_{\text{pattern}}^{(l)} \in \mathbb{R}^D$. Notably, it is imperative to exercise caution when incorporating $\mathbf{h}_{\text{pattern}}^{(l)}$ into the decoder, otherwise the model may inadvertently collapse into an identity mapping function by simply duplicating the input. Therefore, we compress $\mathbf{h}_{\text{pattern}}^{(l)}$ into a probability vector by:

$$\pi = \text{Softmax}(\text{MLP}(\mathbf{h}_{\text{pattern}}^{(l)})), \tag{36}$$

where $\pi = \{\pi_1, \pi_2, \ldots, \pi_k, \ldots, \pi_K\}$ and $\pi_k$ is the probability of choosing the $k$-th memory. To ensure that each memory space receives sufficient training, we do not perform hard selection. Instead, we obtain the final memory through a weighted sum:

$$\mathbf{m}_{\text{final}} = \sum_{k=1}^{K} \pi_k \mathbf{m}_k'. \tag{37}$$

Ultimately, we can leverage the acquired memory to facilitate the decoding process and effectively capture the intricate one-to-many relationship between sequences and patterns.

