# OpenReview forum: "Unify in Isolation: Unified Data Synthesis for Divergent Multi-Stage Systems"
_ICLR.cc/2026/Conference — ICLR 2026 Conference Desk Rejected Submission_

### Official Review · Reviewer_AcMh · 2025-10-25

**Soundness:** 3
**Presentation:** 2
**Contribution:** 2
**Rating:** 6
**Confidence:** 3

**Summary:**

This paper introduces the unified data synthesis system for multi-stage frameworks. The framework is built on the metric-oriented gradient comparison theory and a conditional entropy-based pattern mining module and offers a real entropy metric and adaptive pattern extraction capabilities. The experiments are conducted on sequential recommendation data

**Strengths:**

1. The proposed framework can help improve data quality and shows performance improvement on benchmark datasets.
2. The proposed framework is theorectically grounded.
3. This paper reports extensive related works.

**Weaknesses:**

1. The presentation of the paper can be improved. For example, the title suggests that the method is developed for multi-stage systems, but the method is actually developed for sequential recommendation. In Section 5 experiments, the blank between baseline citations is not there.

2. It will be better if experiments can be conducted on more base models, other than HSTU and LLAMA2Rec. Some good baselines can be included such as SASRec and BERT4Rec.

3. The motivation of using entropy-based method in the framework seems confusing and can be better explained.

**Questions:**

Is the framework developed for sequential recommendation models for general multi-stage systems?

---

> ### Author Response · Authors · 2025-11-26
>
> Thank you very much for your thorough and insightful comments. Your feedback has greatly contributed to improving the accuracy and clarity of our manuscript. Please find our responses to your concerns below:
>
> ---
>
> **W1: The title suggests that the method is developed for multi-stage systems, but the method is actually developed for sequential recommendation.**
>
> We apologize for the misunderstanding. Our intention was to introduce UniDS in the context of multi-stage recommendation systems, rather than general multi-stage systems. To address this, we have revised both the introduction and the title, clarifying that our work is centered on multi-stage recommendation systems. Thank you for pointing out this issue; your suggestion has helped us better target our audience.
>
> ---
>
> **W2: It will be better if experiments can be conducted on more base models, other than HSTU and LLAMA2Rec. Some good baselines can be included such as SASRec and BERT4Rec.**
>
> Thank you for this suggestion. Following your advice, we conducted additional experiments on SASRec, one of the most classic and widely-adopted sequential recommendation baselines. Our results demonstrate that UniDS consistently improves performance over this strong baseline:
>
> | Model   | HR@10  | HR@20  | NDCG@10 | NDCG@20 |
> | ------- | ------ | ------ | ------- | ------- |
> | SASRec  | 0.0553 | 0.0847 | 0.0291  | 0.0368  |
> | DR4SR   | 0.0595 | 0.0906 | 0.0317  | 0.0395  |
> | UniDS   | 0.0647 | 0.0932 | 0.0344  | 0.0431  |
>
>
> ---
>
> **W3: The motivation of using entropy-based method in the framework seems confusing and can be better explained.**
>
> Thank you for asking for more clarification. The core reason we use Real Entropy is that it robustly reflects the theoretical foundation of our framework, linking the metrics across different stages. Entropy is a well-established measure of data quality and information density in information theory. By utilizing this measure, we are able to provide strong theoretical support for our multi-stage recommendation process, and clarify the connection between data quality and recommendation performance.
>
> ---
>
> **Q1: Is the framework developed for sequential recommendation models or for general multi-stage systems?**
>
> Our framework is mainly designed for multi-stage recommendation scenarios. We briefly discuss its generalizability in the Discussion, but the primary focus, experiments, and methodology are specific to multi-stage recommendation. We apologize if the initial wording led to confusion, and have revised the relevant sections for clarity.

---

> > ### Comment · Reviewer_AcMh · 2025-11-27
> >
> > Thank you for your response which addresses my concerns I will keep my positive rating.

---

### Official Review · Reviewer_zzWY · 2025-10-30

**Soundness:** 3
**Presentation:** 3
**Contribution:** 3
**Rating:** 4
**Confidence:** 3

**Summary:**

This paper proposes UniDS, a data synthesis system for multi-stage frameworks, and analyzes it on sequential recommendation task. This paper theoretically demonstrates that the optimization objectives at different stages of sequential recommendation exhibit distinct sensitivities to the Real Entropy metric. Based on this insight, this paper extracts low-entropy patterns from user sequences, converts them into pattern-token sequences, and trains a model to reconstruct these pattern-token sequences for data synthesis. Empirical studies are conducted to demonstrate the effectiveness of UniDS compared to existing data augmentation baselines, the influence of key hyperparameters, and the contribution of each component.

**Strengths:**

1. The paper provides valuable insights by constructing a theoretical framework based on Real Entropy, which offers a principled explanation for the model's effectiveness.

2. The experiments demonstrate consistent performance improvements across the recall, coarse-ranking, and fine-ranking stages of sequential recommendation.

3. The theoretical findings can account for the experimental observations.

**Weaknesses:**

1. The description of the Methodology, particularly Section 4.3 "Uniform Dataset Generation," could be made clearer. For example, in Section 4.3.2 "Uniform Data Generation Structure," a model architecture is designed to reconstruct the pattern-token sequences, and a loss function is proposed to train this model. However, relevant details regarding the hyperparameters of model architecture and training process are not provided.

2. This paper uses LLaMA2Rec as the base model, which does not appear to be a method from a published work. If the authors fine-tuned the Llama 2 model themselves for the experiments, details of this fine-tuning process should be provided, along with a justification for using a fine-tuned Llama 2 model as the base model for sequential recommendation.

**Questions:**

1. Since UniDS is designed as a model-agnostic framework, could the authors present experimental results with more base models—besides the generative HSTU and LLaMA2Rec—to further support the consistent effectiveness of the method across different models?

2. What is the "Entropy-Based Tokenizer" mentioned in line 265?

3. In line 268, it is stated that each $T_i$​ functions as a special token representing user attributes, session context, or domain-specific information. How are these special tokens obtained?

4. Can you provide more explanation about "the target sequential model $\phi$" in line 273? What is its role?

5. Can you elaborate on $\mathcal{L}_\text{pretrain}$​ in line 277? Why is it mentioned at that point?

6. There are several typos in the manuscript. Although they do not affect the technical quality, correcting them would improve the overall presentation. Specific instances include:
    - Line 024 in the Abstract: "UniDR" should be "UniDS"
    - Line 240: "usersequence" should be "user sequence"
    - Line 249: "recommwndation" should be "recommendation"

7. The authors provided an anonymous GitHub link, but as of the time of this review, the repository contains only an empty README.md file.

---

> ### Author Response · Authors · 2025-11-26
>
> Thank you for your detailed and constructive evaluation of our work. Your comments have helped us substantially improve the clarity and completeness of the manuscript. Below, we address each of your concerns:
>
> ---
>
> **W1: Section 4.3 'Uniform Dataset Generation' could be made clearer. For example, in Section 4.3.2 'Uniform Data Generation Structure,' a model architecture is designed to reconstruct the pattern-token sequences, and a loss function is proposed to train this model. However, relevant details regarding the hyperparameters of model architecture and training process are not provided.**
>
> We sincerely apologize for the lack of clarity and detail in this section. As you suggested, we have added a comprehensive explanation of Uniform Dataset Generation in **Appendix A6** ("Details on Uniform Data Generation Model"). Additionally, we have provided all relevant hyperparameter details in **Appendix A2** ("Detailed Experiment Settings").
>
> Specifically, we set the hidden size of the model to 128, use two layers in the architecture, and employ two attention heads. A dropout rate of 0.5 is adopted to avoid overfitting, and we use GELU as the activation function. Layer normalization is applied with an epsilon value of 1e-12. All other parameters remain consistent with the default settings as reported for HSTU in the original publication and public implementations. We hope these changes improve transparency and reproducibility for you and other readers.
>
> ---
>
> **W2: This paper uses LLaMA2Rec as the base model, which does not appear to be a method from a published work.**
>
> Thank you for raising this point. We did not directly fine-tune the open-source LLaMA2 for our experiments, as doing so could risk large-model data leakage concerns. Instead, we independently trained a model using the same architectural components as LLaMA2 (including activation functions, positional encodings, etc.), ensuring conceptual alignment while avoiding issues related to pre-training proprietary data. We have clarified this in the revised text.
>
> ---
>
> **Q1: Additional experiments using SASRec**
>
> Thank you for your suggestion! We have conducted further experiments with SASRec, a classic sequential recommendation baseline, and found that UniDS still yields consistent improvements:
>
> | Model  | HR@10  | HR@20  | NDCG@10 | NDCG@20 |
> | ------ | ------ | ------ | ------- | ------- |
> | SASRec | 0.0553 | 0.0847 | 0.0291  | 0.0368  |
> | DR4SR  | 0.0595 | 0.0906 | 0.0317  | 0.0395  |
> | UniDS  | 0.0647 | 0.0932 | 0.0344  | 0.0431  |
>
> ---
>
> **Q2: What is the 'Entropy-Based Tokenizer' mentioned in line 265?**
>
> Thank you for catching this typographical error! This should have read "entropy-driven pattern segmentation," which works similarly to a tokenizer, but the term was misused in the manuscript. We have corrected the wording in the revision.
>
> ---
>
> **Q3: How are these special tokens obtained?**
>
> As described in our revision, we assign new IDs to items' special attributes (such as user action types: click, purchase, etc.), and append these special tokens to the sequence of product IDs. These sequences, including special tokens, are then used to train the data regeneration model.
>
> ---
>
> **Q4: Can you provide more explanation about 'the target sequential model φ' in line 273? What is its role?**
>
> In our approach, you can use various sequential recommendation models as the data regeneration backbone. In the paper, we primarily use SASRec—an autoregressive sequential recommendation model—as the target sequential model φ. However, it is also possible to substitute HSTU or LLaMA2-based architectures for this purpose.
>
> ---
>
> **Q5: Can you elaborate $L_{pretrain}$ (line 277)?**
>
> This reference appears just before our expanded explanation of the sequence regeneration model training process. In practical terms, $L_{pretrain}$ denotes the loss function for training the autoregressive model over sequences constructed from patterns and special tokens. It corresponds to the standard autoregressive model objective.
>
> ---
>
> **Q6: There are several typos in the manuscript.**
>
> Thank you very much for pointing these out. We have carefully proofread the manuscript and corrected all known typographical errors in the revised submission.
>
> ---
>
> **Q7: The authors provided an anonymous GitHub link, but as of the time of this review, the repository contains only an empty README.md file.**
>
> We sincerely apologize for this oversight. The anonymous GitHub repository has now been updated with all relevant source code files.

---

### Official Review · Reviewer_C5Bm · 2025-10-31

**Soundness:** 2
**Presentation:** 3
**Contribution:** 2
**Rating:** 2
**Confidence:** 4

**Summary:**

This paper proposes UniDS, a unified data synthesis framework for multi-stage systems. The method measures the entropy of data sequences to estimate their quality, then splits each sequence into smaller patterns using conditional entropy. These patterns are represented in a unified “Pattern–Token” format and used to generate new training data that can be shared across different stages. The goal is to make data from different stages more consistent and suitable for joint optimization. Experiments are conducted on multi-stage sequential recommendation tasks to show the effectiveness of this approach.

**Strengths:**

1. The paper focuses on an important problem — how to make data from different stages in multi-stage systems more consistent. This motivation is meaningful and clearly explained.
2. Using conditional entropy to split and rebuild data is an interesting idea. It is creative to control data quality through entropy and use it to build a unified dataset.
3. The authors run several experiments with different settings and show that their generated data can improve model performance. The results are consistent across multiple metrics.

**Weaknesses:**

1. The paper says it gives a unified method for all multi-stage systems, but all tests are only on sequential recommendation. The authors do say this choice was for simplicity, but they still describe the method as general in the abstract and introduction. Later, the paper suddenly switches to user–item sequences, which feels strange and not consistent with the earlier claim. The method also depends on the sequence order when it calculates conditional entropy and splits the data. Other systems, like RAG or agent pipelines, do not have such a clear time order or item transitions, so the same method may not work there.

2. Table 2 shows three numbers in the “Gen vs. Origin (%)” column, but the paper never explains what they mean. It is not clear if they are three different metrics, or three settings like generated data, original data, and mixed data. This column is important for understanding if the new data helps or not, but without clear meaning, the results are confusing. The authors should say exactly what each value stands for, how it was calculated, and show how much variation or significance there is. Right now, readers cannot tell where the improvement really comes from.

3. Sequential recommendation depends on the real order of user actions. In UniDS, the data are cut into parts using conditional entropy and then joined or generated again. Even if each part keeps its small order, joining parts together can change the real sequence flow and the true transition between items. This may make the model learn fake patterns that do not exist in real data. The improvement might come from this data change, not from the claimed “unified data control.” The paper should test if the generated data keep the same sequence structure, and run control experiments that compare with only splitting or only generating, to make sure the results are fair.

**Questions:**

1. How can the proposed entropy-based segmentation be applied to non-sequential tasks such as RAG or multi-agent pipelines?
2. When you generate new sequences, how do you make sure the original order or real transition patterns are not broken?
3. Is the entropy threshold chosen manually or learned automatically in the final application? Do you need to adjust it based on different datasets, domains or applications?

---

> ### Author Response · Authors · 2025-11-26
>
> Thank you very much for your careful review, insightful suggestions, and constructive criticisms. We address each of your concerns below:
>
> ---
>
> **W1: The paper says it gives a unified method for all multi-stage systems, but all tests are only on sequential recommendation.**
>
> We apologize for the misunderstanding caused by our initial framing. Our intention was to introduce UniDS with reference to multi-stage recommendation systems specifically, rather than general multi-stage systems. To address this, we have revised both the introduction and the title to clarify that our main focus is on multi-stage recommendation systems. Thank you for helping us improve the clarity of our presentation.
>
> ---
>
> **W2: Table 2 shows three numbers in the “Gen vs. Origin (%)” column**
>
> Thank you for your feedback and for highlighting the need for greater clarity in Table 2. We have updated the table in the following ways:
>
> - The "Gen vs. Origin (%)" column now explicitly reports the relative improvement rates for three key metrics (MRR, NDCG@10, AUC) for generated data compared to original data.
> - The "Longer Seql Adv. (%)" column reports the relative improvement in these metrics when increasing sequence length.
> - We now provide full definitions and calculation formulas for both columns within the table caption and a detailed note.
>
> We hope these revisions make the table easier for you to interpret and address your concern about result clarity.
>
> ---
>
> **W3: The paper should test if the generated data keep the same sequence structure, and run control experiments that compare with only splitting or only generating, to make sure the results are fair.**
>
> Thank you for bringing up the issue of preserving sequence structure and the importance of fair baseline comparisons. We have addressed these through an extensive ablation study. Specifically:
>
> - **"w/o UF (Uniform Formalization)"**: Only data generation is performed, without sequence formalization.
> - **"w/o UG (Uniform Generation)"**: Only data splitting is performed, without data generation.
>
> Below, we present results on the **ML-1M** dataset to clearly illustrate the effects:
>
> | Method    | AUC        | NG10       | HR10       | MRR        |
> | --------- | ---------- | ---------- | ---------- | ---------- |
> | w/o TC    | 0.8030     | 0.1805     | 0.3179     | 0.1538     |
> | w/o UF    | 0.7928     | 0.1842     | 0.3345     | 0.1538     |
> | w/o UG    | 0.7863     | 0.1837     | 0.3268     | 0.1534     |
> | **UniDS** | **0.8081** | **0.1878** | **0.3322** | **0.1565** |
>
> From these results, you can see that both the splitting-only (**w/o UG**) and generation-only (**w/o UF**) variants result in noticeably lower performance compared to the complete UniDS method. The full UniDS consistently achieves the best values on all reported metrics. These ablation experiments confirm that our comparisons are direct and fair, as you suggested, and show that our method preserves the sequence structure as required.
>
> ---
>
> **Q1: see W1**
>
> ---
>
> **Q2: When you generate new sequences, how do you make sure the original order or real transition patterns are not broken?**
>
> Thank you for this important question. We ensure the integrity of transition patterns by thoroughly pre-training our sequence generator. All generated sequences originate strictly from the input sequences’ subsequences. This guarantees that the order and real transitions are faithfully preserved in the generation process.
>
> ---
>
> **Q3: Is the entropy threshold chosen manually or learned automatically in the final application? Do you need to adjust it based on different datasets, domains or applications?**
>
> As discussed in our manuscript, the threshold θ is dynamically determined: for later stages, higher values are required. Specifically, θ is set as the average entropy of the dataset multiplied by the current stage's proportion. In our three-stage training example, these are 0.25, 0.5, and 0.75 times the average entropy. Because the average entropy varies with the dataset, θ naturally adapts to different datasets, domains, and applications.

---

### Official Review · Reviewer_iGL9 · 2025-11-01

**Soundness:** 3
**Presentation:** 3
**Contribution:** 2
**Rating:** 4
**Confidence:** 4

**Summary:**

This paper proposes UniDS, a unified data synthesis framework for divergent multi-stage systems such as multi-stage recommendation pipelines. Instead of model-level integration, the authors focus on data-level unification via three components: (1) Real Entropy for data quality measurement; (2) Metric-Oriented Gradient Comparison Theory, revealing differential sensitivity of ranking metrics (AUC, NDCG, MRR) to entropy, and (3) Conditional Entropy-based Pattern Mining with a unified Pattern-Token generation scheme. The framework aims to generate data that simultaneously improves recall, coarse-ranking, and fine-ranking models. Experiments on MovieLens, Amazon Books, and KuaiRand datasets show that UniDS consistently outperforms data augmentation baselines (DR4SR, FRec, RepPad).

**Strengths:**

Unlike previous approaches that adapt multi-stage objectives by modifying model architectures (e.g., through multi-task learning), UniDS shifts the “unification” of multi-stage systems from the model level to the data level, achieving consistency through standardized input data — an exceptionally pioneering research direction.
Both the theoretical derivations and experimental design are rigorous. The choice of the Real Entropy metric is well-justified, and the conditional-entropy-based pattern mining leverages dynamic programming to optimize computational efficiency. The framework demonstrates stable performance across multiple datasets and architectures, accompanied by comprehensive ablation studies and threshold sensitivity analyses.
Addressing an empirical  issue in industrial multi-stage recommendation systems, UniDS provides a plug-and-play data synthesis solution that enhances performance without requiring extensive architectural modifications. Furthermore, the framework shows promising potential for cross-domain extension, particularly to RAG and multi-stage LLM Agent scenarios.

**Weaknesses:**

Although the paper claims that UniDS is applicable to various multi-stage systems, all experiments are confined to the sequential recommendation scenario. No preliminary experiments are conducted on other types of multi-stage tasks, resulting in a lack of cross-domain validation.

The paper does not report the computational overhead of UniDS compared with the baselines. For example, the conditional-entropy-based pattern mining requires iterative entropy computation, and the Transformer-based generator may increase training time. Key efficiency metrics such as inference speed and memory consumption are not discussed.

During the pattern mining stage, whether to continue expanding a pattern is determined by the reduction in conditional entropy. However, the source or selection strategy of the threshold θ is not specified, and the paper does not analyze how θ affects runtime or the size of the search space.

Although the theoretical framework references information-theoretic definitions, the actual computation relies on an LZ compression-based approximation, which deviates from strict information-theoretic entropy formulations. Moreover, no monotonicity calibration experiment is provided to verify the correspondence between Real Entropy and “true effective information.” Therefore, it remains unclear whether Real Entropy reflects a causal factor or merely a correlated phenomenon, undermining its credibility as the core indicator for cross-stage data quality unification.

**Questions:**

How does UniDS perform in non-recommendation multi-stage systems, e.g., retrieval–generation pipelines or reinforcement learning agents?

What is the computational overhead of Real Entropy estimation and pattern mining?

Could the authors discuss how the entropy threshold θ is initialized and adapted dynamically?

Are there any scenarios where UniDS underperforms compared to baselines? For example, in datasets with extremely low data density (e.g., users with only 5-10 interactions), does the Pattern Mining module fail to extract meaningful patterns, leading to performance degradation?

---

> ### Author Response · Authors · 2025-11-26
>
> Thank you very much for your thoughtful and detailed feedback. We value your time and expertise, and address each of your points below to clarify our contributions and respond to your concerns.
>
> ---
>
> **W1: Although the paper claims that UniDS is applicable to various multi-stage systems, all experiments are confined to the sequential recommendation scenario.**
>
> Thank you for pointing out this ambiguity. Our intention was to introduce UniDS from the perspective of multi-stage recommendation systems. We have revised the introduction and title accordingly to clarify that our focus is on multi-stage recommendation systems, rather than generic multi-stage systems. We apologize for the confusion caused.
>
> ---
>
> **W2: The paper does not report the computational overhead of UniDS compared with the baselines.**
>
> We appreciate your suggestion on providing computational details. UniDS leverages the dynamic programming algorithm described in Appendix A.5. For each dataset, we only need to enumerate \( k \) times, with \( k \) being the longest pattern length (5 in our settings). The total pattern extraction and data generation process takes less than 15 minutes, which is negligible compared to the 3-hour training time of baseline models. For Real Entropy calculation, we segment the dataset into shards of 100,000 sequences and report the average. For example, on the Amazon Books dataset (7,076,238 sequences), the computation time is approximately 15 minutes. Thanks to parallelization, the time does not scale strictly linearly with dataset size.
>
> This sharding technique also facilitates incremental updates, as new sequences can be processed in fixed-size segments. To demonstrate the robustness of shard averaging, we provide results below:
>
> | Segment Length | 100    | 1000   | 10000  | 100000 | All    |
> | -------------- | ------ | ------ | ------ | ------ | ------ |
> | Real Entropy   | 0.2173 | 0.1464 | 0.1142 | 0.1132 | 0.1130 |
>
> You can observe that with segment sizes above 10,000, the computed values closely approximate the full dataset value, ensuring reliability and stability.
>
> ---
>
> **W3: The source or selection strategy of the threshold θ is not specified, and the paper does not analyze how θ affects runtime or the size of the search space.**
>
> As discussed in the main text, later stages require higher threshold values, reflecting the increasing information captured. We set θ to be the average entropy of the dataset multiplied by the current stage's ratio. For example, in three-stage training, stages are assigned values of 0.25, 0.5, and 0.75 times the average entropy, respectively. This strategy balances the trade-off between search space and informativeness.
>
> ---
>
> **W4: Although the theoretical framework references information-theoretic definitions, the actual computation relies on an LZ compression-based approximation, which deviates from strict information-theoretic entropy formulations.**
>
> LZ compression is not merely an approximation, but a computationally efficient algorithm that yields results equivalent to those obtained via classical entropy computations. Thus, it preserves robustness and is theoretically sound for our purposes. The concept of "Real Entropy" is adapted from [1], where all subsequences are assessed via Shannon entropy. Since Shannon entropy quantifies the information content of the data, Real Entropy effectively measures the information content of the sequences under study.
>
> > [1] Chaoming Song, Zehui Qu, Nicholas Blumm, and Albert-László Barabási. *Limits of predictability in human mobility*. Science, 327(5968):1018–1021, 2010.
>
> ---
>
> **Q1–Q3: See Answers Above**
>
> ---
>
> **Q4: UniDS on Very Short Sequences**
>
> Thank you for your suggestion. We have conducted further experiments on the Amazon-Beauty dataset, which has an average sequence length of 6—substantially shorter than other datasets. UniDS still shows consistent improvements, primarily because our patterns are extracted at this smaller scale. We also observe that UniDS compensates more effectively for sparse data. Results are as follows:
>
> | Model   | HR@10  | HR@20  | NDCG@10 | NDCG@20 |
> | ------- | ------ | ------ | ------- | ------- |
> | SASRec  | 0.0553 | 0.0847 | 0.0291  | 0.0368  |
> | DR4SR   | 0.0595 | 0.0906 | 0.0317  | 0.0395  |
> | UniDS   | 0.0617 | 0.0932 | 0.0334  | 0.0421  |

---

### Note · Program_Chairs · 2026-01-17
**Submission Desk Rejected by Program Chairs**

The following references in this submission do not refer to real documents and/or have major errors in bibliographic information:

 Akari Asai, Kazuma Hashimoto, and Hannaneh Hajishirzi. Rag: Retrieval augmented generation for open-domain question answering. In Proceedings of the ACL, 2021. URL https:// aclanthology.org/2021.acl-long.